# Multi-level Certified Defense Against Poisoning Attacks in Offline Reinforcement Learning

**Shijie Liu**[1][*], **Andrew C. Cullen**[1], **Paul Montague**[2], **Sarah Erfani**[1], **Benjamin I. P. Rubinstein**[1]
[1]School of Computing and Information Systems, University of Melbourne, Melbourne, Australia
[2]Defence Science and Technology Group, Adelaide, Australia
[*]shijie3@unimelb.edu.au

## Abstract

Similar to other machine learning frameworks, Offline Reinforcement Learning (RL) is shown to be vulnerable to poisoning attacks, due to its reliance on externally sourced datasets, a vulnerability that is exacerbated by its sequential nature. To mitigate the risks posed by RL poisoning, we extend certified defenses to provide larger guarantees against adversarial manipulation, ensuring robustness for both per-state actions, and the overall expected cumulative reward. Our approach leverages properties of Differential Privacy, in a manner that allows this work to span both continuous and discrete spaces, as well as stochastic and deterministic environments—significantly expanding the scope and applicability of achievable guarantees. Empirical evaluations demonstrate that our approach ensures the performance drops to no more than $50\%$ with up to $7\%$ of the training data poisoned, significantly improving over the $0.008\%$ in prior work (Wu et al., 2022), while producing certified radii that is 5 times larger as well. This highlights the potential of our framework to enhance safety and reliability in offline RL.

## 1 Introduction

Offline Reinforcement Learning (RL), also known as batch RL, involves training policies entirely from pre-collected datasets. Doing so is particularly advantageous in scenarios where directly interacting with the environment is costly, risky, or infeasible, such as healthcare (Wang et al., 2018), autonomous driving (Pan et al., 2017), and robotics (Gürtler et al., 2023). Due to this, offline RL mechanistically shares the same vulnerability to *data poisoning attacks* (Kiourti et al., 2020; Wang et al., 2021) as traditional classifiers, in which adversarial manipulation of the training data can lead to suboptimal or harmful decisions. Such vulnerability is further intensified by the dependence on external datasets collected by unknown behavioral agents and the dynamic, sequential decision-making process of RL. Across industrial users of RL, poisoning attacks are broadly considered to pose the most pressing security risk (Kumar et al., 2020), with offline settings being of particular concern (Zhang et al., 2021a). These intrinsic risks highlight the need for specialized defensive strategies to be developed, in order to support RL deployments.

Defenses in RL share many of the same risks as defenses deployed for other Machine Learning paradigms, in that they can be circumvented by a motivated attacker. By contrast, *certified defenses* offer theoretical guarantees of robustness against worst-case adversarial manipulations. Such robustness guarantees are particularly desirable in safety-critical domains and have been extensively explored in classification tasks (Lecuyer et al., 2019; Salman et al., 2019; Cullen et al., 2022). However the direct applicability of these techniques to RL is made more challenging due to the complex sequential dependency and interactive nature of RL (Kiourti et al., 2020).

While some works (Ye et al., 2023; Zhang et al., 2021a; Lykouris et al., 2023) have established robustness bounds for RL from a *theoretical* perspective, they typically rely on significant simplifications of the problem setting that do not reflect the complexities of real-world RL scenarios, limiting their applicability. Furthermore, these robustness bounds are typically expressed in terms

of the optimality gap to the practically unattainable Bellman optimal policy, offering qualitative insights rather than quantitative certification value of the robustness.

To circumvent these limitations, recent research has begun to consider how practical certifications can be constructed, resulting in the COPA approach (Wu et al., 2022). This approach provides computable lower bounds on *cumulative reward* and *certified radii*, to ensure policies are robust against poisoning attacks. However, despite its utility, COPA is fundamentally limited to discrete action spaces and deterministic settings, which constrains it to a small subset of potential RL environments. Additionally, it certifies only individual trajectories without offering robustness guarantees for the overall performance of the learned policy.

To resolve these limitations, within this work, we propose the **Mu**lti-level **C**ertified **D**efenses (MuCD) against poisoning attacks in general offline RL settings, offering multi-level robustness guarantees across different levels of poisoning. To assist in this, we distinguish between adversarial attacks against RL that involve *trajectory-level* poisoning, which occur during the data collection process; and *transition-level* poisoning, which occurs after the training dataset is collected. In response to the existence of these threat models, we propose employing certifications that employ *action-level* robustness (expanding upon Wu et al. 2022) to ensure that critical states are safeguarded against being entered; and *policy-level* robustness, which provides a lower bound on the expected cumulative reward. This latter framework naturally aligns with RL policy's primary goal (Prudencio et al., 2024). To achieve these certifications, our framework comprises two stages: a Differential Privacy (DP) based randomized training process and robustness certification methods. These certifications are broadly applicable to RL, covering both discrete and continuous action spaces, as well as deterministic and stochastic environments.

Our contributions on both a theoretical and empirical level are:

- Formulating both multi-level attacks and certifications for poisoning attacks in offline RL, enabling comprehensive analysis of the robustness of the offline RL training process.
- Proposing the first practical certified defense framework that provides computable robustness certification in terms of both *per-state action stability* and *expected cumulative reward* in *general offline RL settings*.
- Experimentally demonstrating significant improvements over past certification frameworks across varying environments and RL algorithms.

## 2 RELATED WORKS

**Offline and Online RL.** RL approaches can be broadly categorized as online or offline learning. Of these, online learning algorithms involve agents learning by interacting with the environment in real-time, driving advances in a range of fields (Silver et al., 2017; Schulman et al., 2017; Kendall et al., 2019). While frameworks like Policy Gradient (Sutton et al., 1999) and Actor-Critic (Mnih, 2016) can effectively learn policies for online RL, their trial-and-error exploration may lead to unintended harmful outcomes and unsafe decisions during the learning process, which is of particular concern to safety-critical areas such as healthcare (Yu et al., 2021) and finance (Nevmyvaka et al., 2006). By contrast, offline (or batch) RL is considered safer, as it learns to emulate pre-collected data to create optimal policies without interacting with the environment (Lange et al., 2012). Algorithms such as Deep Q-Network (DQN) (Mnih et al., 2013), Implicit Q-Learning (IQL) (Kostrikov et al., 2021) and C51 (Bellemare et al., 2017) have demonstrated effective in leveraging historical data to optimize decision-making without further exploration.

**Poisoning Attacks in Offline RL.** Adversarial attacks are a well-documented threat to machine learning, where motivated adversaries manipulate models to induce unexpected behaviors. Among these, poisoning attacks (Barreno et al., 2006; Biggio et al., 2013)—which deliberately corrupt the training data to degrade the performance of learned models—are particularly concerning for offline RL, due to its reliance on pre-collected datasets and the complex dynamics of RL frameworks (Kumar et al., 2020; Kiourti et al., 2020). Adversaries can target specific components of the data, such as in reward poisoning attacks (Wu et al., 2023), or broadly corrupt the entire dataset as in general poisoning attacks (Wang et al., 2021). Corruption can occur after data has been collected cleanly (Zhang et al., 2021a) or during data collection (Ye et al., 2023; Gong et al., 2024).

**Certified Defenses.** In response to these attacks, a range of defensive mechanisms have been proposed. Of these, certified defenses have drawn particular interest, due to their ability to provide *robustness* guarantees against attack existence for classification tasks (Peri et al., 2020; Lecuyer et al., 2019; Liu et al., 2021; Cullen et al., 2024b), however applying these methods to RL directly has proven challenging due to RL's sequential dependency (Kiourti et al., 2020). While some research has addressed certified robust RL in the context of reward poisoning (Banihashem et al., 2021; Nika et al., 2023), extensions to general poisoning attacks have been more limited, with results primarily restricted to *theoretical analyses* of simplified variants (using linear MDPs or assuming bounded distances to Bellman optimality) of offline RL under attack occurring during data collection (Ye et al., 2023) or afterward (Yang et al., 2024; Zhang et al., 2021a). Crucially, these robustness approaches typically produce bounds expressed in asymptotic measures of the optimality gap between the learned policy and the theoretical optimal policy, which have limited practical applicability.

By contrast, COPA (Wu et al., 2022) recently demonstrated that per-state certification for RL could be computed by adapting the Deep Partition Aggregation (DPA) (Levine & Feizi, 2021) method from classification tasks. Based on that, they proposed a tree-search approach that exhaustively explores all possible trajectories to compute a lower bound on the cumulative reward. However, such an approach intrinsically limits it to discrete action spaces and deterministic environments. Consequently, their certification framework only applies to specific, repeatable trajectories and fails to provide robustness guarantees for the reward or policy in more general scenarios.

## 3 PRELIMINARIES

In this section, we formulate the offline RL framework as an episodic finite-horizon Markov Decision Process (MDP), establishing the foundation for our discussion. We then outline the dataset construction process and introduce a comprehensive multi-level poisoning attack model to address potential risks in offline RL training, along with the objectives for certified defense. Finally, we highlight key concepts from Differential Privacy (DP) that underpin our approach.

### 3.1 MULTI-LEVEL POISONING

**Framework.** The RL framework is modeled as an episodic finite-horizon MDP, represented by the tuple $(\mathcal{S}, \mathcal{A}, P, R, H, \gamma)$, where $\mathcal{S}$ is the state space, $\mathcal{A}$ is the action space, $P : \mathcal{S} \times \mathcal{A} \to \Delta(\mathcal{S})$ is the stochastic transition function with $\Delta(\cdot)$ defining the set of probability measures, $R : \mathcal{S} \times \mathcal{A} \to \mathbb{R}$ is the bounded reward function, $H$ is the time horizon, and $\gamma \in \mathbb{R}$ is the discount factor.

At a time step $t$, an RL agent in state $s_t \in \mathcal{S}$ selects an action $a_t = \pi(s_t)$ according to its policy $\pi \in \Pi : \mathcal{S} \to \mathcal{A}$. Upon executing $a_t$, the agent transitions to the subsequent state $s_{t+1} \sim P(s_t, a_t)$ and receives a reward $r_t = R(s_t, a_t)$. The tuple $(s_t, a_t, s_{t+1}, r_t)$ is referred to as a transition, and the sequence of transitions $\{(s_t, a_t, s_{t+1}, r_t)\}_{t=0}^{H-1}$ over one episode constitutes a trajectory $\tau$.

**Offline RL Datasets.** Offline RL employs a dataset $D = \{\tau_j\}_{j=1}^{M}$ consisting of $M$ trajectories, or equivalently $N$ transitions $D = \{(s_i, a_i, s_{i+1}, r_i)\}_{i=1}^{N}$, collected by an unknown behavioral policy $\pi_\beta$. The agent learns its policy $\pi$ from this dataset without further interaction with the environment, which provides opportunities for an adversary to poison the training data during the behavioral policy execution or after the collection process.

**Trajectory-level Poisoning.** Trajectory-level poisoning occurs when the adversary corrupts the data collection process. The adversary, with full knowledge of the MDP, can observe all the historical transitions and alter transitions $\{(s_t, a_t, s_{t+1}, r_t)\}$ at arbitrarily many time steps by replacing them with $\{(\tilde{s}_t, \tilde{a}_t, \tilde{s}_{t+1}, \tilde{r}_t)\}$. As any alteration can affect subsequent transitions and propagate through gameplay, the following definition quantifies corruption by the number of modified *trajectories*, following the adversarial models in robust statistics (Diakonikolas et al., 2019) and robust RL (Zhang et al., 2021b; Wu et al., 2022) settings.

**Definition 3.1** (Trajectory-level poisoning). *Assume an adversary can make up to $r$ changes, including additions, deletions, or alterations, to the trajectories within a clean dataset $D = \{\tau_j\}_{j=1}^{M}$. Then the set of all possible poisoned datasets is $\mathcal{B}_{trj}(D, r) := \{\tilde{D} : |D \ominus_{trj} \tilde{D}| \leq r\}$, where $|D \ominus_{tra} \tilde{D}|$ measures the minimum number of changes to the trajectories required to map $D$ to $\tilde{D}$.*

**Transition-level Poisoning.** In transition-level poisoning, the adversary modifies transitions after the data is collected. Therefore, alterations are limited to the specific transitions $\{(\tilde{s}_i, \tilde{a}_i, \tilde{s}_{i+1}, \tilde{r}_i)\}_{i=1}^r$ that were directly modified, without impacting any subsequent transitions across the trajectory. Hence, we quantify the corruption by the total number of modified *transitions* through the following definition.

**Definition 3.2** (Transition-level poisoning). *Assume an adversary can make up to $r$ changes, including additions, deletions, or alterations, to the transitions within a clean dataset $D = \{(s_i, a_i, s_{i+1}, r_i)\}_{i=1}^N$. Then the set of all possible poisoned datasets is $\mathcal{B}_{tra}(D, r) := \{\tilde{D} : |D \ominus_{tra} \tilde{D}| \leq r\}$, where $|D \ominus_{tra} \tilde{D}|$ measures the minimum number of changes to the transitions required to map $D$ to $\tilde{D}$.*

## 3.2 MULTI-LEVEL ROBUSTNESS CERTIFICATION

To ensure robustness against data poisoning in RL, we aim to certify *test-time* performance of a policy $\pi = \mathcal{M}(D)$ trained on a clean dataset $D$ with the training algorithm $\mathcal{M}$. In doing so, we will bound the difference between $\pi$ and the equivalent $\tilde{\pi} = \mathcal{M}(\tilde{D})$ trained upon a poisoned dataset $\tilde{D}$, subject to a constraint on the difference between $D$ and $\tilde{D}$.

**Policy-level Robustness.** Offline RL algorithms aim to find an optimal policy that maximizes the expected cumulative reward for all trajectories induced by the policy (Prudencio et al., 2024). Thus we first aim to construct certifications regarding the *expected cumulative reward*. The expected cumulative reward is denoted as $J(\pi) = \mathbb{E}_{\sigma, \xi} \left[ \sum_t \gamma^t r_t \mid \pi \right]$, where $\xi$ represents the randomness of the environment and $\sigma$ represents the randomness introduced by the training algorithm. The following definition demonstrates how policy-level robustness certifications can construct a lower bound on the expected cumulative reward, henceforth labelled as $\underline{J}_r$, under a poisoning attack of size $r$.

**Definition 3.3** (Policy-level robustness certification). *Given a clean dataset $D$, a policy-level certification ensures that a policy $\tilde{\pi} = \mathcal{M}(\tilde{D})$ trained on any poisoned dataset $\tilde{D} \in \mathcal{B}(D, r)$ will produce an expected cumulative reward $J(\tilde{\pi}) \geq \underline{J}_r$ with probability at least $1 - \delta$.*

**Action-level Robustness.** Beyond ensuring generalised robustness, it is crucial to be able to guarantee the safety of the agent by ensuring it avoids catastrophic outcomes and entering undesirable states (Gu et al., 2024). Therefore, we also aim to certify the stability of the agent's actions on a per-state basis during testing. The action-level robustness certification in a discrete action space at state $s_t$ under a poisoning attack of size $r$ is defined in the following definition.

**Definition 3.4** (Action-level robustness certification). *Given a clean dataset $D$ and state $s_t$, the action-level robustness certification states that for any poisoned dataset $\tilde{D} \in \mathcal{B}(D, r)$, the clean and poisoned policies produce the same action $\pi(s_t) = \tilde{\pi}(s_t)$ where $\pi = \mathcal{M}(D)$ and $\tilde{\pi} = \mathcal{M}(\tilde{D})$, with a probability of at least $1 - \delta$.*

## 3.3 DIFFERENTIAL PRIVACY

DP quantifies privacy loss when releasing aggregate statistics or trained models on sensitive data (Dwork et al., 2006; Abadi et al., 2016; Friedman & Schuster, 2010). As DP can be used to measure the sensitivity of outputs to input perturbations, it is well aligned to use in certifications, leading to it being employed in multiple works (Lecuyer et al., 2019; Ma et al., 2019; Cullen et al., 2024a). The remainder of this section will introduce key properties of DP as employed by our work, with more detailed explanations of the *Approximate-DP* (ADP) and *Rényi-DP* (RDP) mechanisms deferred to Appendix A.1.

Our work relies upon two key principles of DP—the *post-processing property*, that any computation applied to the output of a DP algorithm preserves the same DP guarantee (Dwork et al., 2006); and the *outcomes guarantee* (Liu et al., 2023; Mironov, 2017), as explained in the following definition specifically for ADP and RDP.

**Definition 3.5** (Outcomes guarantee for ADP and RDP). *A randomised function $\mathcal{M}$ is said to preserve a $(\mathcal{K}, r)$-outcomes guarantee if for any function $K \in \mathcal{K}$ such that for all datasets $D_1$ and*

$D_2 \in \mathcal{B}(D_1, r)$, and for all measurable output sets $S \subseteq \text{Range}(\mathcal{M})$ if

$$\Pr[\mathcal{M}(D_1) \in S] \leq \text{K}(\Pr[\mathcal{M}(D_2) \in S]) \ . \tag{1}$$

In ADP, the function family $\mathcal{K}$ is parameterized by $\epsilon, \delta$ as $\mathcal{K}_{\epsilon,\delta}(x) = e^\epsilon x + \delta$, while in RDP $\mathcal{K}$ is parameterized by $\epsilon, \alpha$ as $\mathcal{K}_{\epsilon,\alpha}(x) = (e^\epsilon x)^{\frac{\alpha-1}{\alpha}}$.

## 4 APPROACH

Our novel certified defense employs a DP-based *randomized training process* and provides two unique certification methods to construct both *action-level* and *policy-level* robustness certification against *transition* and *trajectory* level poisoning attacks.

### 4.1 RANDOMIZED TRAINING PROCESS

Our certification requires the training algorithm $\mathcal{M}$ to ensure the DP guarantee of its output policy $\pi$ with respect to the training dataset $D$. DP mechanisms introduce randomness into the training process by adding calibrated noise in updating the parameters, producing a randomized policy $\pi$. Empirically, this can be represented as a set of $p$ policy instances $(\hat{\pi}_1, \cdots, \hat{\pi}_p)$. As each instance undergoes the same training process, this can be easily parallelized for efficiency. For larger datasets, further efficiency gains can be achieved by training each instance on a subset $D_{sub} \subseteq D$.

Our specific approach employs the Sampled Gaussian Mechanism (SGM) (Mironov et al., 2019) to ensure DP guarantee at the transition-level $\mathcal{B}_{tra}$, and adapts the DP-FEDAVG (McMahan et al., 2017) for the trajectory-level $\mathcal{B}_{trj}$ DP guarantee. Details of the training algorithms are deferred to Appendix A.2. For the remainder of this paper, $\mathcal{B}$ will represent either $\mathcal{B}_{tra}$ or $\mathcal{B}_{trj}$, depending on whether the applied DP training algorithm provides transition- or trajectory-level guarantees.

### 4.2 POLICY-LEVEL ROBUSTNESS CERTIFICATION

Consider a DP training algorithm $\mathcal{M}$ (as described in Section 4.1) that preserves a $(\mathcal{K}, r)$-outcomes guarantee for the clean dataset $D$, producing the clean policy $\pi = \mathcal{M}(D)$. When the dataset is poisoned as $\tilde{D}$, the resulting policy is denoted as $\tilde{\pi} = \mathcal{M}(\tilde{D})$. To certify the policy-level robustness as in Definition 3.3 in terms of the lower bound of expected cumulative reward, we denote the testing time expected cumulative reward of a policy $\pi$ as expressed by

$$J(\pi) = \underset{\sigma}{\mathbb{E}}\left[C(\pi)\right] \quad \text{where} \quad C(\pi) = \underset{\xi}{\mathbb{E}}\left[\sum_{t=0}^{H-1} \gamma^t r_t | \pi\right] \ , \tag{2}$$

that $\sigma$ represents the training randomness, and $\xi$ represents the environment randomness.

To provide bounds over the expected output of the DP mechanisms, we propose the following lemma that extends the outcomes guarantee from probability to the expected value.

**Lemma 4.1** (Expected Outcomes Guarantee for ADP and RDP). *If an $\mathcal{M}$ that produces bounded outputs in $[0, b], b \in \mathbb{R}^+$ satisfies $(\mathcal{K}, r)$-outcomes guarantee, then for any $\tilde{D} \in \mathcal{B}(D, r)$ the expected value of the outputs of the $\mathcal{M}$ must satisfy: If $\mathcal{K}$ denotes the function family of ADP $\mathcal{K}_{\epsilon,\delta}$,*

$$e^{-\epsilon}(\mathbb{E}[\mathcal{M}(D)] - b\delta) \leq \mathbb{E}[\mathcal{M}(\tilde{D})] \leq e^\epsilon \mathbb{E}[\mathcal{M}(D)] + b\delta \ . \tag{3}$$

*Similarly, if $\mathcal{K}$ denotes the function family of RDP $\mathcal{K}_{\epsilon,\alpha}$,*

$$e^{-\epsilon}(b^{-1/\alpha}\mathbb{E}[\mathcal{M}(D)])^{\frac{\alpha}{\alpha-1}} \leq \mathbb{E}[\mathcal{M}(\tilde{D})] \leq b^{1/\alpha}(e^\epsilon \mathbb{E}[\mathcal{M}(D)])^{(\alpha-1)/\alpha} \ , \tag{4}$$

*where the expectation is taken over the randomness in $\mathcal{M}$.*

*Proof.* While a full proof is contained within Appendix A.3, here we present an informative sketch. The upper bound of the expected value can be obtained by integrating over the right-tail distribution function of the probabilities in Equation (1) by Fubini's Theorem (Fubini, 1907). The integral results of ADP and RDP can be derived by respectively employing Lecuyer et al. (2019) and Hölder's Inequality. The lower bound follows by the symmetry in the roles of $D_1, D_2$ in DP, and by $K$ being strictly monotonic. □

With this preliminary result, we now turn to the main result of this section, which is to establish that DP learning algorithms ensure policy-level robustness against poisoning attacks up to size $r$, with an extension to real-valued cumulative rewards deferred to Appendix A.4.

**Theorem 4.2** (Policy-level robustness by outcomes guarantee). *Consider an RL environment with bounded cumulative reward in the range $[0, b], b \in \mathbb{R}^+$, as well as a randomized offline RL policy $\pi = \mathcal{M}(D)$ constructed by the learning algorithm $\mathcal{M}$ using training dataset $D$. If $\mathcal{M}$ preserves a ADP $(\mathcal{K}, r)$-outcomes guarantee, then each $K \in \mathcal{K}_{\epsilon, \delta}$ with corresponding $\epsilon, \delta$ satisfies the policy-level robustness of size $r$ for any poisoned dataset $\tilde{D} \in \mathcal{B}(D, r)$ as*

$$J(\tilde{\pi}) \geq \underline{J}_r(\tilde{\pi}) = e^{-\epsilon}(J(\pi) - b\delta) \ . \tag{5}$$

*If $\mathcal{M}$ preserves a RDP $(\mathcal{K}, r)$-outcomes guarantee, then each $K \in \mathcal{K}_{\epsilon, \alpha}$ with corresponding $\epsilon, \alpha$ satisfies the policy-level robustness of size $r$ as*

$$J(\tilde{\pi}) \geq \underline{J}_r(\tilde{\pi}) = e^{-\epsilon}(b^{-1/\alpha}J(\pi))^{\frac{\alpha}{\alpha-1}} \ . \tag{6}$$

*Proof.* This result is a direct consequence of the $(\mathcal{K}, r)$-outcomes guarantee and the post-processing property, as $C(\tilde{\pi}) = C(\mathcal{M}(\tilde{D}))$ is a post-computation applied to the output of the DP mechanism $\mathcal{M}$, hence it satisfies the same $(\mathcal{K}, r)$-outcomes guarantee by the post-processing property. By Lemma 4.1, the expected value $J(\tilde{\pi}) = \mathbb{E}[C(\tilde{\pi})]$ and $J(\pi) = \mathbb{E}[C(\pi)]$ satisfy the inequality in Equation (5) and Equation (6) for ADP and RDP respectively. $\qquad \square$

To compute the policy-level robustness using Theorem 4.2, we need to obtain the lower bound of $J(\pi)$ to substitute into the Equation (5) and Equation (6). Consider the cumulative reward of the policy $\pi$ as a random variable $X = \sum_t^H \gamma^t r_t$ where $J(\pi) = \mathbb{E}_{\sigma, \xi}[X]$. The estimations of $X$ can be obtained by playing the games $m$ times using the trained policy instances $(\hat{\pi}_1, \cdots, \hat{\pi}_p)$, and obtain its empirical Cumulative Distribution Function (CDF) $\hat{F}_X(x)$. By the Dvoretzky–Kiefer–Wolfowitz inequality (Dvoretzky et al., 1956), the true CDF $F_X(x)$ must be bounded by an empirical CDF $\hat{F}_X(x)$ of a finite sample size $m$ with probability at least $1 - \delta$ as

$$\hat{F}_X(x) - \varepsilon \leq F_X(x) \leq \hat{F}_X(x) + \varepsilon \quad \text{where} \quad \varepsilon = \sqrt{\frac{\ln \frac{2}{\delta}}{2m}} \ . \tag{7}$$

The expected value of the random variable $X$ in the bounded range $[0, b]$ can be expressed by the true CDF $F_X(x)$ and bounded by the empirical CDF $\hat{F}_X(x)$ as

$$J(\pi) = \mathbb{E}_{\sigma, \xi}[X] = \int_0^b (1 - F_X(x)) \, dx \geq \int_0^b (1 - (\hat{F}_X(x) - \varepsilon)) \, dx \ , \tag{8}$$

and thus allows us to construct the lower bound $\underline{J}_r(\tilde{\pi})$, as required for policy-level certification.

## 4.3 ACTION-LEVEL ROBUSTNESS CERTIFICATION

In this section, we propose a method for certifying action-level robustness as in Definition 3.4 in terms of the stability of output actions. To achieve this, we begin by considering the decision-making process of a policy $\pi$ given state $s_t$ in a discrete action space $\mathcal{A} = \{A_1, \cdots, A_L\}$. For each instance $\hat{\pi}_i$ of the policy, the action $a_{t,i}$ is selected based on the highest *action-value* $a_{t,i} = \arg\max_{a_l \in \mathcal{A}} Q_{\hat{\pi}_i}(s_t, a_l) = \mathbb{E}_{\tilde{\pi}_i} \left[ \sum_{t=0}^{H-1} \gamma^t r_t \mid s_0 = s_t, a_0 = a_l \right]$. Without loss of generality, we denote the action $a_t$ chosen by the randomized policy $\pi$ as the one with the highest *inferred scores* $I_{A_l}(s_t, \pi)$, where $\sum_{A_l \in \mathcal{A}} I_{A_l}(s_t, \pi) = 1$ and $I_{A_l}(s_t, \pi) \in [0, 1]$. The inferred score function of the randomized policy $\pi$ takes the form

$$I_{A_l}(s_t, \pi) = \Pr[\arg\max_{a_i} Q_\pi(s_t, a_i) = A_l] \ , \tag{9}$$

indicating that the action is induced as the most likely one, which can be estimated unbiasedly by using a majority vote among all policy instances. We then propose the following lemma, which extends the outcome guarantees to inferred scores.

**Lemma 4.3** (Inferred scores outcomes guarantee). *If $\mathcal{M}$ preserves a $(\mathcal{K}, r)$-outcomes guarantee for a dataset $D$, and there exist an $I$ that maps the learned policy $\pi = \mathcal{M}(D)$ and a state $s_t$ to an inferred score, then for any $K \in \mathcal{K}$, it must hold that for any action $A_l \in \mathcal{A}$ and any policy $\tilde{\pi} = \mathcal{M}(\tilde{D})$ trained with dataset $\tilde{D} \in \mathcal{B}(D, r)$:*

$$K^{-1}(I_{A_l}(s_t, \pi)) \leq I_{A_l}(s_t, \tilde{\pi}) \leq K(I_{A_l}(s_t, \pi)) \ . \tag{10}$$

*Proof.* The composition $I \circ \mathcal{M}$ preserves the same outcomes guaranteed by the post-processing property. The reverse inequality is derived from the symmetry in the roles of the datasets in DP, with $K$ being strictly monotonic. The outcomes guarantee can be directly converted to Equation (10) by defining $S = \{\pi : \arg\max_{a_i} Q_\pi(s_t, a_i) = A_l\}$. ☐

With the bounds over inferred scores by outcomes guarantee, we present the theorem that specifies the conditions under which a DP learning algorithm maintains action-level robustness against poisoning attacks of size $r$ in any arbitrary state $s_t$.

**Theorem 4.4** (Action-level robustness by outcomes guarantee). *Consider an offline RL training dataset $D$, a randomized learning algorithm $\mathcal{M}$ that satisfies a $(\mathcal{K}, r)$-outcomes guarantee and outputs a policy $\pi = \mathcal{M}(D)$. Let $I$ be the inferred score function, and $s_t$ be an arbitrary test-time input state with the corresponding output action $a_t = \arg\max_{a_l \in \mathcal{A}} I_{a_l}(s_t, \pi)$. If there exist $K_1, K_2 \in \mathcal{K}$ such that:*

$$K_1^{-1}(I_{a_t}(s_t, \pi)) > \max_{a_l \in \mathcal{A} \setminus \{a_l\}} K_2(I_{a_l}(s_t, \pi)) \tag{11}$$

*then the algorithm preserves action-level robustness at state $s_t$ under a poisoning attack of size $r$.*

*Proof.* As the policy selects the action that maximises the inferred score, the objective of certifying action-level robustness is equivalent to proving that the inferred score of $a_t$ is larger than the inferred score of any other actions $a_l \in \mathcal{A} \setminus \{a_t\}$ for any poisoned dataset $\tilde{D} \in \mathcal{B}(D, r)$, as

$$\forall \tilde{D} \in \mathcal{B}(D, r)$$
$$I_{a_t}(s_t, \mathcal{M}(\tilde{D})) > \max_{a_l \in \mathcal{A} \setminus \{a_l\}} I_{a_l}(s_t, \mathcal{M}(\tilde{D})) \ . \tag{12}$$

Given $\mathcal{M}$ preserves a $(\mathcal{K}, r)$-outcomes guarantee, then for any $K_1, K_2 \in \mathcal{K}$, the following inequalities can be derived by Lemma 4.3 as

$$I_{a_t}(s_t, \mathcal{M}(\tilde{D})) > K_1^{-1}(I_{a_t}(s_t, \pi))$$
$$\max_{a_l \in \mathcal{A} \setminus \{a_l\}} I_{a_l}(s_t, \mathcal{M}(\tilde{D})) < \max_{a_l \in \mathcal{A} \setminus \{a_l\}} K_2(I_{a_l}(s_t, \pi)) \ . \tag{13}$$

Therefore, if there exists $K_1$ and $K_2$ that satisfy the condition in Equation (11), the transitive property of inequalities ensures that the condition in Equation (12) is also satisfied. ☐

The maximum tolerable poisoning size $r_t$ can be calculated while maintaining action-level robustness by way of the policy instances $(\hat{\pi}_1, \cdots, \hat{\pi}_p)$ and the condition in Theorem 4.4. At each time, the selected action $a_t$ is that with the highest inferred score across the policy instances, with upper and lower bounds estimated simultaneously through sampling the outputs of the policy instances to a confidence level of at least $1 - \delta$ by the SIMUEM method (Jia et al., 2020). We substitute the lower bound of $I_{a_t}(s_t, \pi)$ and the upper bound of $\max_{a_t \in \mathcal{A} \setminus \{a_t\}} I_{a_{t2}}(s_t, \pi)$ into the condition of Equation (11). By Theorem 4.4 we can then certify whether action-level robustness is achieved at state $s_t$ under a poisoning size $r$. The maximum tolerable poisoning size $r_t$ is determined through a binary search over the domain of $\mathcal{K}$ to find the $K_1$ and $K_2$ that satisfies the condition for maximising $r$. We defer the details of this process to Appendix A.5.

## 5 EXPERIMENTS

In this section, we evaluate our proposed certified defenses under scenarios of either transition- or trajectory-level poisoning (Section 3.1) for policy-level and action-level robustness (Section 3.2). To

facilitate these, we conducted evaluations using Farama Gymnasium (Towers et al., 2023) discrete Atari games Freeway and Breakout, as well as the continuous action space Mujoco game Half Cheetah. We also employed the D4RL (Fu et al., 2020) dataset and the Opacus (Yousefpour et al., 2021) DP framework. Our environments were trained using DQN (Mnih et al., 2013), IQL (Kostrikov et al., 2021) and C51 (Bellemare et al., 2017), implemented with Convolutional Neural Networks (CNN) in PyTorch on a NVIDIA 80GB A100 GPU. Further results considering the robustness of these defenses to empirical attacks are presented within Appendix A.7.

Our offline RL datasets consist of 2 million transitions for each game, with corresponding trajectory counts of 976 for Freeway, 3,648 for Breakout, and 2,000 for Half Cheetah. In all experiments, the sample rates $q$ in the DP training algorithms were adjusted to achieve a batch size of 32, with varying noise multipliers $\sigma$ as detailed in the results. Uncertainties were estimated within a confidence interval suitable for $\delta = 0.001$. For each game, the number of policy instances $p$, as described in Section 4.1, is set to 50. The number of estimations of expected cumulative reward $m$ is set to 500, with 10 estimations per policy instance, as detailed in Section 4.2.

## 5.1 ACTION-LEVEL ROBUSTNESS RESULTS

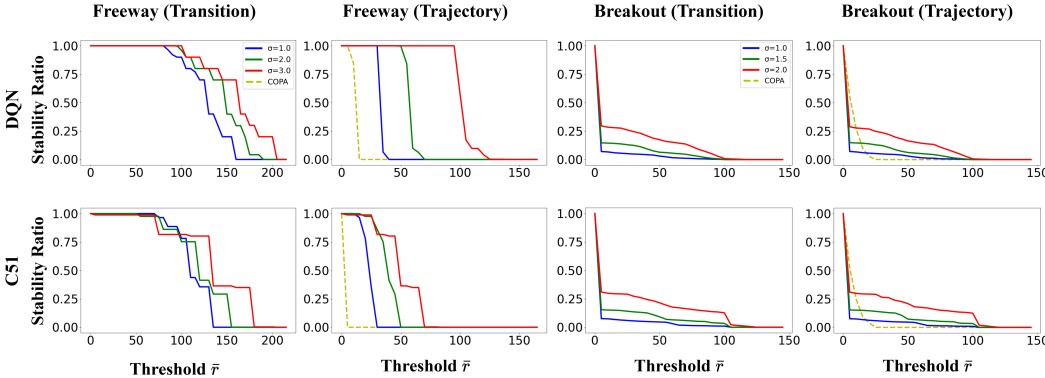

Figure 1: Stability ratio against the tolerable poisoning threshold $\bar{r}$ for *action-level robustness* using DQN and C51 for the Freeway and Breakout environments under transition- or trajectory-level poisoning attacks. Blue, Green and Red lines represent different noise levels $\sigma$ during the randomized training process as $\sigma = \{1, 2, 3\}$ for Freeway and $\{1, 1.5, 2\}$ for Breakout, while the yellow dashed line denotes COPA, which can only be calculated for trajectory-level poisoning.

| Environment | Method | Noise | Avg. Cumulative Reward | | Action-level Mean Radii | | | |
|---|---|---|---|---|---|---|---|---|
| | | | | | DQN | | C51 | |
| | | | DQN | C51 | Transition | Trajectory | Transition | Trajectory |
| Freeway | Proposed (RDP) | 0.0 | 20.1 | 21.3 | N/A | N/A | N/A | N/A |
| | | 1.0 | 16.9 | 16.1 | 128.1 | 32.6 | 111.6 | 22.1 |
| | | 2.0 | 16.6 | 15.3 | 145.5 | 58.7 | 119.3 | 37.8 |
| | | 3.0 | 16.0 | 15.1 | 160.0 | 102.4 | 134.5 | 49.7 |
| | COPA | N/A | 16.4 | 16.4 | N/A | 10.1 | N/A | 9.7 |
| Breakout | Proposed (RDP) | 0.0 | 385.4 | 389.3 | N/A | N/A | N/A | N/A |
| | | 1.0 | 366.6 | 369.0 | 3.4 | 3.2 | 4.0 | 3.9 |
| | | 1.5 | 320.8 | 270.4 | 7.9 | 7.6 | 9.5 | 9.3 |
| | | 2.0 | 268.4 | 102.7 | 17.7 | 16.9 | 22.0 | 21.7 |
| | COPA | N/A | 325.7 | 330.1 | N/A | 6.6 | N/A | 6.3 |

Table 1: Testing time average cumulative reward in a clean environment and the mean of maximum tolerable poisoning size $r_t$ of the action-level robustness.

We will now evaluate action-level robustness across varying RL algorithms and environments for RDP, with additional ADP based experiments and statistics provided in Appendix A.6. This analysis considers the **mean** and **maximum** value of $r_t$ across a set of evaluated trajectories as well the

**stability ratio** (Wu et al., 2022). This latter metric represents the proportion of time steps in a trajectory where the maximum tolerable poisoning size $r_t$ of action-level robustness is maintained under a poisoning attack of size up to a given threshold $\bar{r}$ for a trajectory length $H$ by way of

$$\text{Stability Ratio} = \frac{1}{H} \sum_{t=0}^{H-1} \mathbb{1}[r_t \geq \bar{r}] \ . \tag{14}$$

To interpret our results, it is important to emphasise that a higher stability ratio at larger thresholds signifies better certified robustness. Models trained with higher noise achieve stronger certified robustness, at the cost of clean performance decreasing in terms of the average cumulative rewards, as shown in Table 1. In concert with Figure 1 it is clear that DQN produces a consistently higher certifications than C51 in Freeway, while matching performance in Breakout. These differences arise from DQNs ability to adapt to noisey training process, allowing it to ameliorate the impact of these perturbations without a significant drop in performance. It is also important to note that the Freeway consistently shows better action-level robustness than Breakout, suggesting that Freeway supports more stable and robust policies.

Given the effectively interchangeable action-level robustness of the variants as reported in COPA (Wu et al., 2022), our comparisons are constructed against the basic PARL variant in the same setting, with the number of partitions set to 50. For a fair comparison, the models trained at a noise level represented by the green line in each game achieve comparable performance to COPA in terms of Avg. Cumulative Reward, as shown in Table 1. Our technique's maximum tolerable poisoning size—indicated by the x-axis intersection in Figure 1—is approximately 5 times larger than other approaches, which confirms that our approach produces stronger robustness in states where certain actions are highly preferable, typically during critical moments. Additionally, our method achieves a higher mean value of the tolerable poisoning size across all steps as shown in Table 1, demonstrating better certification for most states.

## 5.2 POLICY-LEVEL ROBUSTNESS RESULTS

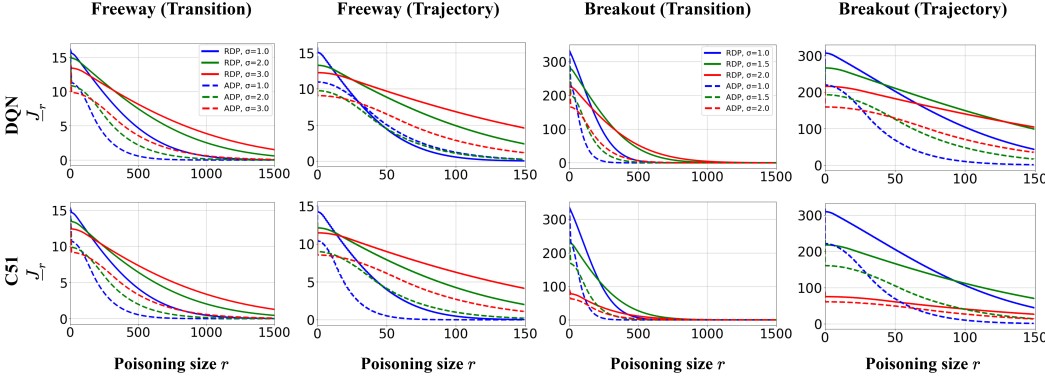

Figure 2: Policy-level robustness certifications, capturing the lower bound of the expected cumulative reward $\underline{J}_r$ against poisoning size $r$ for Atari games. Solid and dashed lines represent RDP and ADP derived guarantees respectively, with colors indicating noise levels as per Figure 1.

To assess policy-level robustness, we turn to the measure $\underline{J}_r(\pi)$, which directly reflects the policy-level robustness of the learned policy against a poisoning attack of size $\bar{r}$ (as in Definition 3.3), with Figure 2 showing this for the discrete games Freeway and Breakout. We also consider the continuous game Half Cheetah as shown in Figure 3, where the benign training yielded an average cumulative reward of 96.47 and randomized training with noise levels set at $\sigma = 1.0$, 2.0, and 3.0 yielded average cumulative rewards of 90.5, 87.0, and 83.4, respectively. The results can be compared from different aspects. In terms of the poisoning level, for the same policy certification $\underline{J}_r$, the tolerable poisoning size $r$ in transition-level certification is about 10 times larger than in trajectory-level. Given that each trajectory consists of approximately 1,000 transitions, the total

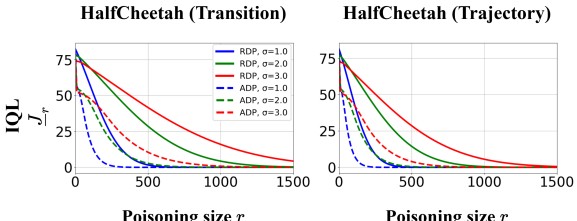

Figure 3: Policy-level robustness certification for the continuous action game Mujoco Half Cheetah, using RL algorithm IQL. The plot is formulated in the same way as Figure 2.

number of transitions that can be altered in trajectory-level certification is actually higher than in transition-level. This aligns with the nature of the threat models: in trajectory-level poisoning, not all transition changes are assumed adversarial, whereas in transition-level poisoning, the adversary can more precisely target key transitions to minimize the number of modifications.

In analyzing the influence of RL algorithms and certification methods, DQN consistently demonstrates higher robustness certification than C51, which is consistent with the analysis from the action-level certification. RDP's tight quantification of privacy loss, particularly in handling iterative function composition in deep networks, provides a significant advantage over ADP in all settings.

Lastly, our approach accommodates more general RL settings, in that it can be applied to both discrete and continuous action spaces, as well as deterministic and stochastic environments, and significantly improves upon the performance of extant techniques, namely COPA. As COPA certifies the cumulative reward for specific trajectories rather than the expected cumulative reward of the policy, our comparison with COPA is often implicit. The limited applicable scenarios of COPA stem from its reliance upon exhaustive tree search in certifying cumulative reward, which is fundamentally incompatible with environments involving randomness or continuous action spaces, and limiting the trajectory length to $400$ in Freeway and $75$ in Breakout due to the exponential growth of the tree size, while the default trajectory lengths are $2,000$ and $600$, respectively. As a result, the maximum cumulative reward COPA can certify is restricted to $5$ in Freeway and $2$ in Breakout. By contrast, our approach has no such limitations regarding environment settings or trajectory length. Furthermore, for Freeway, COPA only allows $0.008\%$ of trajectories in the training dataset to be poisoned while certifying less than a $50\%$ performance drop, whereas our method achieves a much higher ratio of $7.17\%$. We observe a similar delta in relative performance within the Breakout games, where COPA's ratio of $0.0075\%$ is significantly smaller than $2.05\%$ observed for our approach.

## 6 CONCLUSIONS

This work explored how certified defenses against poisoning attacks can be both constructed and enhanced in offline RL. To do this, we introduced a novel framework that leverages Differential Privacy mechanisms to provide the first practical certified defense in a general offline RL setting. While past works have only considered theoretical robustness bounds or are limited to specific RL settings, our framework is able to offer both action-level stability and policy-level lower bounds with respect to the expected cumulative reward of the learned policy. Furthermore, our experiments across a wide range of RL environments and algorithms demonstrate robust certifications in practical applications, significantly outperforming other state-of-the-art defense frameworks.

Our work suggests several potential directions for future research. First, developing a unified DP training algorithm that simultaneously supports both transition- and trajectory-level certified defenses could significantly enhance robustness. Additionally, defense performance may be further improved by designing a more sophisticated noise injection mechanism that adapts noise levels dynamically, rather than uniform noise throughout the entire training process.

## 7 ACKNOWLEDGEMENTS

This research was supported by The University of Melbourne's Research Computing Services and the Petascale Campus Initiative. This work was also supported in part by the Australian Department of Defence through the Defence Science and Technology Group (DSTG). Sarah Erfani is in part supported by the Australian Research Council (ARC) Discovery Early Career Researcher Award (DECRA) DE220100680.

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

# A  APPENDIX

## A.1  ADP AND RDP DEFINITIONS

**Definition A.1** (Approximate-DP). *A randomised function $\mathcal{M}$ is said to be $(\epsilon, \delta)$-Approximate DP (ADP) if for all datasets $D_1$ and $D_2$ for which $D_2 \in \mathcal{B}(D_1, 1)$, and for all measurable output sets $S \subseteq \text{Range}(\mathcal{M})$:*

$$\Pr[\mathcal{M}(D_1) \in S] \leq e^\epsilon \Pr[\mathcal{M}(D_2) \in S] + \delta \ , \tag{15}$$

*where $\epsilon > 0$ and $\delta \in [0, 1)$ are chosen parameters.*

ADP with the privacy guarantee as expressed in Equation (15) is the most commonly used format in DP research. Smaller values of the privacy budget $\epsilon$ restrict the (multiplicative) influence of a participant joining dataset $D_2$ to form $D_1$, thereby limiting the probability of any subsequent privacy breach. The confidence parameter $\delta$ relaxes this guarantee by allowing for the possibility that no bound is provided on privacy loss in the case of low-probability events.

To bound the residual risk from ADP, Rényi-DP (RDP) was introduced by Mironov (2017). Rényi-DP provide tighter quantification of privacy through sequences of function composition, as required when iteratively training a deep net on sensitive data, which leads to improved certifications in practice.

**Definition A.2** (Rényi-DP). *A randomised function $\mathcal{M}$ preserves $(\alpha, \epsilon)$-Rényi-DP, with $\alpha > 1, \epsilon > 0$, if for all datasets $D_1$ and $D_2 \in \mathcal{B}(D_1, 1)$:*

$$\mathrm{D}_\alpha \left( \mathcal{M}(D_1) \| \mathcal{M}(D_2) \right) \leq \varepsilon \ , \tag{16}$$

*where $\mathrm{D}_\alpha$ represents the Rényi divergence of finite order $\alpha \neq 1$ between two distributions $P$ and $Q$ defined over the same probability space $\mathcal{X}$ with densities $p$ and $q$ as*

$$\mathrm{D}_\alpha(P \| Q) \triangleq \frac{1}{\alpha - 1} \ln \int_{\mathcal{X}} q(x) \left( \frac{p(x)}{q(x)} \right)^\alpha \mathrm{d}x \ . \tag{17}$$

The generalization to Definition 3.5 incorporates *group privacy* (Dwork et al., 2006) to extend DP to adjacent datasets to pairs datasets that differ in up to $r$ data points $\mathcal{B}(D_1, r)$. The ADP's function family $\mathcal{K}$ is derived directly from the Definition A.1, while the RDP's family is obtained by applying Hölder's inequality to the integral of the density function in the Rényi divergence (Mironov, 2017).

## A.2  DP TRAINING ALGORITHMS

Several works have extended differential privacy to RL by developing privacy-preserving algorithms that balance the trade-off between model performance and privacy guarantees. To tackle the distinct challenges in RL, such as the sequential dependency, multi-sourced data, notable contributions have been made, including techniques for regret minimization RL with privacy guarantee (Vietri et al., 2020; Dann et al., 2017; Garcelon et al., 2021), off-policy evaluation (Balle et al., 2016), and distributional RL (Ono & Takahashi, 2020).

The proposed methods require the training algorithm $\mathcal{M}$ to preserve the DP guarantee of its output policy $\pi$ regarding the training dataset $D$. Depending on the DP training mechanisms, the trained private policy in the context of offline RL can achieve either transition- or trajectory-level DP guarantees, meaning the $\mathcal{B}$ used in the aforementioned DP definitions can be $\mathcal{B}_{tra}$ or $\mathcal{B}_{trj}$ respectively.

**Transition-level DP Training Method SGM.**  While numerous differential privacy (DP) mechanisms have been proposed and extensively studied in machine learning (Abadi et al., 2016; Mironov et al., 2019), most rely on adding noise directly to the training samples. Instead, the Sampled Gaussian Mechanism (SGM) (Mironov et al., 2019) introduces randomness through both noise injection and sub-sampling, providing a better privacy cost. In SGM, each element of the training batch is sampled without replacement with uniform probability $q$ from the training dataset. Additionally, Gaussian noise is added to the gradients during each weight update step. The training algorithm is illustrated in Algorithm 1 When applied to a model $\mathcal{M}$, SGM preserves $(\alpha, \epsilon)$-RDP, where $\epsilon$ is determined by the parameters $(\alpha, \mathcal{M}, q, \sigma)$. This RDP guarantee can be further transformed into $(\epsilon, \delta)$-ADP using the conversion method described by Balle et al. (2019).

---

**Algorithm 1** Sampled Gaussian Mechanism (SGM) for a Model $\mathcal{M}$ using Dataset $D$

---

**Require:** Dataset $D$ with $n$ samples, sampling ratio $q$, noise multiplier $\sigma$, number of iterations $T$, learning rate $\eta$
**Ensure:** Private model $\mathcal{M}$
 1: Initialize model parameters $\theta_0$
 2: **for** $t = 1, 2, \ldots, T$ **do**
 3:     Sample a mini-batch $B_t \subseteq D$ by selecting each element of $D$ with probability $q$ without replacement
 4:     Compute gradients $\nabla\mathcal{L}(\theta_{t-1}; B_t)$ with respect to the mini-batch
 5:     Clip gradients: $\bar{\nabla}\mathcal{L} = \frac{\nabla\mathcal{L}}{\max(1, \frac{\|\nabla\mathcal{L}\|}{C})}$ where $C$ is the clipping threshold
 6:     Add Gaussian noise: $\tilde{\nabla}\mathcal{L} = \bar{\nabla}\mathcal{L} + \mathcal{N}(0, \sigma^2 C^2 I)$
 7:     Update model parameters: $\theta_t = \theta_{t-1} - \eta\tilde{\nabla}\mathcal{L}$
 8: **end for**
 9: **return** Differentially private model $\mathcal{M}$ with parameters $\theta_T$

---

**Trajectory-level DP Training Method DP-FEDAVG.** As demonstrated in the SGM, clipping per-sample gradients makes it unsuitable for trajectory-level DP, where the privacy cost needs to be accounted for on a per-trajectory basis. To address this, we utilize DP-FEDAVG (McMahan et al., 2017), initially designed for client-level privacy in federated learning, which can be adapted to scenarios where training data is naturally segmented, such as trajectory data in offline RL. The core idea of DP-FEDAVG is as follows: at each iteration $t$, a subset $B_t$ of trajectories is sampled from the dataset $D$ with probability $q$ without replacement. A single gradient $\nabla\mathcal{L}(\theta_{t-1}; \tau_t)$ is then computed and clipped with constant $C$ for each trajectory. An unbiased estimator of the average gradient of the subset is then calculated, with sensitivity bounded by the $C$ divided by the batch size. Finally, the Gaussian mechanism is applied with noise magnitude $\sigma$, and the model is updated using the noisy gradient. The details of the training algorithm is shown in Algorithm 2.

---

**Algorithm 2** Model Training with DP-FEDAVG

---

**Require:** Dataset $D$, sampling ratio $q \in (0, 1)$, noise multiplier $\sigma$, clipping norm $C$, local epochs $E$, batch size $B$, learning rate $\eta$
**Ensure:** Private model $\mathcal{M}$
 1: Initialize model parameters $\theta_0$
 2: **for** each iteration $t \in [0, T-1]$ **do**
 3:     $U_t \leftarrow$ (sample with replacement trajectories from $D$ with probability $q$)
 4:     **for** each trajectory $\tau_k \in U_t$ **do**
 5:         Clone current model $\theta_{\text{start}} \leftarrow \theta_t$
 6:         **for** each local epoch $i \in [1, E]$ **do**
 7:             $\mathcal{B} \leftarrow$ (split $\tau$'s data into size $B$ batches)
 8:             **for** each batch $b \in \mathcal{B}$ **do**
 9:                 $\theta \leftarrow \theta - \eta\nabla\mathcal{L}(\theta; b)$
10:                 $\theta \leftarrow \theta_{\text{start}} + \text{PerLayerClip}(\theta - \theta_{\text{start}}; C)$
11:             **end for**
12:         **end for**
13:         $\Delta_{t,k}^{\text{clipped}} \leftarrow \theta - \theta_{\text{start}}$
14:     **end for**
15:     $\Delta_t^{\text{avg}} \leftarrow \frac{\sum_{k \in U_t} \Delta_{t,k}^{\text{clipped}}}{qK}$
16:     $\tilde{\Delta}_t^{\text{avg}} \leftarrow \Delta_t^{\text{avg}} + \mathcal{N}\left(0, \left(\frac{\sigma C}{qK}\right)^2\right)$
17:     $\theta_{t+1} \leftarrow \theta_t + \tilde{\Delta}_t^{\text{avg}}$
18: **end for**

---

In addition to the DP training algorithms discussed above, it is worth highlighting the existence of more advanced DP algorithms capable of offering tighter privacy guarantees, enhanced computational efficiency, or optimality analysis (Gopi et al., 2021; Geng & Viswanath, 2015). These advancements enable more precise privacy bounds and contribute to further reinforcing the robustness

of the certification for our proposed defense mechanism. However, it is crucial to emphasize that the primary focus of our work is on establishing a general framework for integrating DP into certified defenses for offline RL. This framework is designed to be adaptable, allowing for the incorporation of more advanced DP mechanisms in future developments.

### A.3 Proof of the Expected Outcomes Guarantee

**Lemma A.1** (Expected Outcomes Guarantee for ADP and RDP). *Suppose a randomized function $\mathcal{M}$, with bounded output $[0, b], b \in \mathbb{R}^+$, satisfies $(\mathcal{K}, r)$-outcomes guarantee. Then for any $\tilde{D} \in \mathcal{B}(D, r)$, if $\mathcal{K}$ denotes the function family of ADP $\mathcal{K}_{\epsilon,\delta}$, the expected value of its outputs satisfies:*

$$e^{-\epsilon}(\mathbb{E}[\mathcal{M}(D)] - b\delta) \leq \mathbb{E}[\mathcal{M}(\tilde{D})] \leq e^{\epsilon}\mathbb{E}[\mathcal{M}(D)] + b\delta \ , \tag{18}$$

*if $\mathcal{K}$ denotes the function family of RDP $\mathcal{K}_{\epsilon,\alpha}$, the expected value of its outputs satisfies:*

$$e^{-\epsilon}(b^{-1/\alpha}\mathbb{E}[\mathcal{M}(D)])^{\frac{\alpha}{\alpha-1}} \leq \mathbb{E}[\mathcal{M}(\tilde{D})] \leq b^{1/\alpha}(e^{\epsilon}\mathbb{E}[\mathcal{M}(D)])^{(\alpha-1)/\alpha} \ , \tag{19}$$

*where the expectation is taken over the randomness in $\mathcal{M}$.*

*Proof.* The expected value can be obtained by integrating over the right-tail distribution function of the probabilities in Equation (1) by Fubini's Theorem (Fubini, 1907) as

$$\mathbb{E}[\mathcal{M}(\tilde{D})] = \int_0^b \Pr[\mathcal{M}(\tilde{D}) > t] \, dt \ . \tag{20}$$

In the case of ADP, for any $K \in \mathcal{K}_{\epsilon,\delta}$ that parameterized by $\epsilon, \delta$, we have

$$\mathbb{E}[\mathcal{M}(\tilde{D})] \leq \int_0^b e^{\epsilon} \Pr[\mathcal{M}(D) > t] + \delta \, dt = e^{\epsilon}\mathbb{E}[\mathcal{M}(D)] + b\delta \ . \tag{21}$$

In the case of RDP, for any $K \in \mathcal{K}_{\epsilon,\alpha}$ that parameterized by $\epsilon, \alpha$, we have

$$\mathbb{E}[\mathcal{M}(\tilde{D})] \leq \int_0^b (e^{\epsilon} \Pr[\mathcal{M}(D) > t])^{(\alpha-1)/\alpha} \, dt \ . \tag{22}$$

Recall Hölder's Inequality, which states that for real-valued functions $f$ and $g$, and real $p, q > 1$, such that $1/p + 1/q = 1$,

$$\|fg\|_1 \leq \|f\|_p \|g\|_q \ . \tag{23}$$

By Hölder's Inequality setting $p = \alpha$ and $q = \alpha/(\alpha-1)$, $f(t) = 1$, $g(t) = \Pr[\mathcal{M}(D) > t]^{(\alpha-1)/\alpha}$, allows for us to state that

$$\begin{aligned}
\mathbb{E}(M(\tilde{D})) &\leq e^{\epsilon(\alpha-1)/\alpha} (\int_0^b 1^{\alpha} dt)^{1/\alpha} (\int_0^b \Pr[\mathcal{M}(D) > t] dt)^{(\alpha-1)/\alpha} \\
&= e^{\epsilon(\alpha-1)/\alpha} b^{1/\alpha} (\mathbb{E}(\mathcal{M}(D)))^{(\alpha-1)/\alpha} \\
&= b^{1/\alpha} (e^{\epsilon}\mathbb{E}(\mathcal{M}(D)))^{(\alpha-1)/\alpha} \ .
\end{aligned} \tag{24}$$

The alternative inequality follows by both $K$ being strictly monotonic and symmetry in the roles of $D_1, D_2$ for DP. □

### A.4 Policy-level Robustness Certification for Real-valued Reward

To certify policy-level robustness in real-valued cumulative reward, we first extend the Lemma 4.1 to the expected value in real number.

**Lemma A.2** (Real-valued Expected Outcomes Guarantee for ADP and RDP). *Suppose a randomized function $\mathcal{M}$, with bounded output $[a, b], a \in \mathbb{R}^-, b \in \mathbb{R}^+$, satisfies $(\mathcal{K}, r)$-outcomes guarantee. Then for any $\tilde{D} \in \mathcal{B}(D, r)$, if $\mathcal{K}$ denotes the function family of ADP $\mathcal{K}_{\epsilon,\delta}$, the expected value of its outputs satisfies:*

$$\mathbb{E}[\mathcal{M}(\tilde{D})] \geq e^{-\epsilon}(\mathbb{E}[\mathcal{M}(D)^+] - b\delta) - (e^{\epsilon}\mathbb{E}[\mathcal{M}(D)^-] - a\delta) \tag{25}$$

$$\mathbb{E}[\mathcal{M}(\tilde{D})] \leq e^{\epsilon}\mathbb{E}[\mathcal{M}(D)^+] + b\delta - e^{-\epsilon}(\mathbb{E}[\mathcal{M}(D)^-] + a\delta) \tag{26}$$

*if $\mathcal{K}$ denotes the function family of RDP $\mathcal{K}_{\epsilon,\alpha}$, the expected value of its outputs satisfies:*

$$\mathbb{E}[\mathcal{M}(\tilde{D})] \geq e^{-\epsilon}(b^{-1/\alpha}\mathbb{E}[\mathcal{M}(D)^+])^{\frac{\alpha}{\alpha-1}} - (-a)^{1/\alpha}(e^\epsilon\mathbb{E}[\mathcal{M}(D)^-])^{(\alpha-1)/\alpha} \tag{27}$$

$$\mathbb{E}[\mathcal{M}(\tilde{D})] \leq b^{1/\alpha}(e^\epsilon\mathbb{E}[\mathcal{M}(D)^+])^{(\alpha-1)/\alpha} - e^{-\epsilon}((-a)^{-1/\alpha}\mathbb{E}[\mathcal{M}(D)^-])^{\frac{\alpha}{\alpha-1}} \tag{28}$$

*where the expectation is taken over the randomness in $\mathcal{M}$, $\mathbb{E}[\mathcal{M}(D)^+]$ represents the expected value of all non-negative $\mathcal{M}(D)$, $\mathbb{E}[\mathcal{M}(D)^-]$ represents the expected value of all negative $\mathcal{M}(D)$.*

*Proof.* We extend the Fubini's Theorem from non-negative values to real values as:

$$\mathbb{E}[X] = \int_0^\infty \Pr[X \geq t]\,dt - \int_{-\infty}^0 \Pr[X \leq t]\,dt \tag{29}$$

which can be derived as

$$\text{Let } X^+ := \begin{cases} X & \text{if } X \geq 0 \\ 0 & \text{otherwise} \end{cases}$$
$$X^- := \begin{cases} -X & \text{if } X < 0 \\ 0 & \text{otherwise} \end{cases} \tag{30}$$

Then, we have

$$X = X^+ - X^- \rightarrow \mathbb{E}[X] = \mathbb{E}[X^+] - \mathbb{E}[X^-]$$
$$\mathbb{E}[X^+] = \int_0^\infty \Pr[X^+ \geq t]dt = \int_0^\infty \Pr[X \geq t]dt \tag{31}$$
$$\mathbb{E}[X^-] = \int_0^\infty \Pr[X^- \geq t]dt = \int_0^\infty \Pr[X \leq -t]dt = \int_{-\infty}^0 \Pr[X \leq t]dt$$

The expected value $\mathbb{E}[\mathcal{M}(\tilde{D})]$ in range $[a,b]$ with $\mathcal{M}$ satisfies $(\mathcal{K},r)$-outcomes guarantee, can be written as

$$\mathbb{E}[\mathcal{M}(\tilde{D})] = \int_0^b \Pr[\mathcal{M}(\tilde{D}) \geq t]dt - \int_a^0 \Pr[\mathcal{M}(\tilde{D}) \leq u)du$$
$$= \int_0^b \Pr[\mathcal{M}(\tilde{D}) \in T)dt - \int_a^0 \Pr[\mathcal{M}(\tilde{D}] \in U)du \tag{32}$$
$$\geq \int_0^b K^{-1}(\Pr[\mathcal{M}(D) \in T])dt - \int_a^0 K(\Pr[\mathcal{M}(D) \in U])du$$

where $K \in \mathcal{K}$. For the cases of ADP and RDP, replace the $K$ with corresponding function as outlined in Definition 3.5. □

Then the Theorem 4.2 can be extended to the case of real-valued expected cumulative reward as,

**Theorem A.3** (Policy-level robustness by outcomes guarantee in real-value range). *Consider an RL environment with bounded cumulative reward in the range $[a,b], a \in \mathbb{R}^-, b \in \mathbb{R}^+$, an offline RL training dataset $D$, and a learning algorithm $\mathcal{M}$ that takes the training dataset $D$ and outputs the randomized policy $\pi = \mathcal{M}(D)$. If $\mathcal{M}$ preserves a $(\mathcal{K},r)$-outcomes guarantee in ADP, then for each $K \in \mathcal{K}_{\epsilon,\delta}$ with corresponding $\epsilon,\delta$ satisfies the policy-level robustness of size $r$ for any poisoned dataset $\tilde{D} \in \mathcal{B}(D,r)$ as*

$$J(\tilde{\pi}) \geq e^{-\epsilon}(J(\pi)^+ - b\delta) - (e^\epsilon J(\pi)^- - a\delta) . \tag{33}$$

*If $\mathcal{M}$ preserves a $(\mathcal{K},r)$-outcomes guarantee in RDP, then for each $K \in \mathcal{K}_{\epsilon,\alpha}$ with corresponding $\epsilon,\alpha$ satisfies the policy-level robustness of size $r$ as*

$$J(\tilde{\pi}) \geq e^{-\epsilon}(b^{-1/\alpha}J(\pi)^+)^{\frac{\alpha}{\alpha-1}} - (-a)^{1/\alpha}(e^\epsilon J(\pi)^-)^{(\alpha-1)/\alpha} , \tag{34}$$

*where $J(\pi)^+$ denotes the expected value of all non-negative cumulative rewards, $J(\pi)^-$ denotes the expected value of all negative cumulative rewards.*

*Proof.* The proof is similar to the proof of Theorem 4.2, instead replace the usage of Lemma 4.1 to Lemma A.2 for the real-valued expected outcomes guarantee. □

## A.5 Additional Details of Action-level Certification

Here we provide additional details of the action-level certification process. As discussed in Section 4.3, the inferred scores are estimated by the sampling from the policy instances $(\hat{\pi}_1, \cdots, \hat{\pi}_p)$. Due to uncertainty, we obtain the upper and lower bounds of the inferred score with a confidence interval of at least $1 - \alpha$ via the SIMUEM method (Jia et al., 2020) based on the Clopper-Pearson method. Specifically, the SIMUEM directly estimates the upper and lower bounds of the inferred scores $I_{A_l}(s_t, \pi) = \Pr[\arg\max_{a_i} Q_\pi(s_t, a_i) = A_l]$ based on the frequencies $(n_i, \cdots, n_L)$ of each $A_l$ produced by the policy instances as

$$\underline{I}_{A_l} = Beta\left(\frac{\alpha}{L}; n_l, p - n_l + 1\right),$$
$$\overline{I}_{A_i} = Beta\left(1 - \frac{\alpha}{L}; n_i + 1, p - n_i\right), \quad \forall i \neq l \ .$$

To determine the maximum tolerable poisoning size $r_t$ for a given state $s_t$, a binary search is performed over the domain of $\mathcal{K}$. As described in Section 3.3 and Appendix A.2, $\mathcal{K}$ represents the set of $(\delta, \epsilon)$ or $(\alpha, \epsilon)$ pairs that satisfy the privacy guarantees of the respective DP training algorithm. The binary search operates within a predefined range, such as $(0, 500)$, aiming to identify the largest radius $r_t$ that meets the condition specified in Theorem 4.4, provided there exist $K_1$ and $K_2$ within the domain of $\mathcal{K}$.

## A.6 Additional Results of Action-level Certification

The Figure 4 and Table 1 show additional experimental results of action-level robustness certification and training statics.

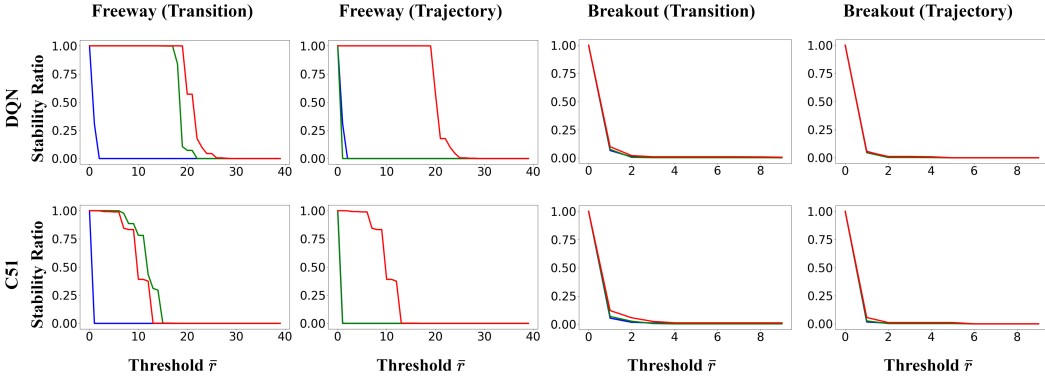

Figure 4: Stability ratio versus the tolerable poisoning threshold $\bar{r}$ for action-level robustness with ADP. Results are presented for two Atari games, Freeway and Breakout with RL algorithms DQN and C51 under transition- and trajectory-level poisoning. The blue, green, and red lines represent our proposed certified defense.

## A.7 Additional Results against Empirical Attacks

To assess the performance of certifications relative to trajectory-level attacks, we implemented two attacks against HalfCheetah when defended using RDP at $\sigma = 2.0$, with the results presented below. These attacks include one in which the rewards in a subset of trajectories are replaced with $r'_i \sim$ Uniform$[-1, 1]$ (Random Reward); and one where they are replaced by $r'_i = -r_i$ (Adversarial Reward), following the methodology of Ye et al. (2023). For each attack, our experiments were conducted over 150 runs to provide the estimated expected cumulative reward (Est. ECR) with 95% confidence intervals, and the minimum cumulative reward among all the runs. The corresponding policy-level robustness certification (certified lower bound on ECR) $\underline{J}_r$ for each poisoning size 10% ($r = 200$) and 20% ($r = 400$) are shown as the same as in Figure 3 in the paper.

| Environment | Method | Noise | Action-level Max Radii | | | |
| --- | --- | --- | --- | --- | --- | --- |
| | | | DQN | | C51 | |
| | | | Transition | Trajectory | Transition | Trajectory |
| Freeway | Proposed (RDP) | 0.0 | N/A | N/A | N/A | N/A |
| | | 1.0 | 159 | 37 | 144 | 29 |
| | | 2.0 | 186 | 66 | 186 | 59 |
| | | 3.0 | 200 | 136 | 200 | 83 |
| | COPA | N/A | N/A | 13 | N/A | 10 |
| Breakout | Proposed (RDP) | 0.0 | N/A | N/A | N/A | N/A |
| | | 1.0 | 99 | 98 | 100 | 100 |
| | | 1.5 | 99 | 99 | 100 | 100 |
| | | 2.0 | 118 | 117 | 120 | 120 |
| | COPA | N/A | N/A | 25 | N/A | 24 |

Table 2: Our proposed method with RDP. The maximum value of maximum tolerable poisoning size $r_t$ of the action-level robustness for the evaluated environments, certified methods, noise levels, and RL algorithms.

| Environment | Method | Noise | Action-level Mean Radii | | | |
| --- | --- | --- | --- | --- | --- | --- |
| | | | DQN | | C51 | |
| | | | Transition | Trajectory | Transition | Trajectory |
| Freeway | Proposed (ADP) | 0.0 | N/A | N/A | N/A | N/A |
| | | 1.0 | 0.3 | 0.3 | 0.0 | 0.0 |
| | | 2.0 | 18.0 | 0.7 | 11.3 | 1.7 |
| | | 3.0 | 20.5 | 20.0 | 9.6 | 2.1 |
| Breakout | Proposed (ADP) | 0.0 | N/A | N/A | N/A | N/A |
| | | 1.0 | 0.1 | 0.07 | 0.1 | 0.04 |
| | | 1.5 | 0.09 | 0.05 | 0.1 | 0.03 |
| | | 2.0 | 0.18 | 0.08 | 0.3 | 0.10 |

Table 3: Our proposed method with ADP. The mean value of maximum tolerable poisoning size $r_t$ of the action-level robustness for the evaluated environments, certified methods, noise levels, and RL algorithms.

| Environment | Method | Noise | Action-level Max Radii | | | |
| --- | --- | --- | --- | --- | --- | --- |
| | | | DQN | | C51 | |
| | | | Transition | Trajectory | Transition | Trajectory |
| Freeway | Proposed (ADP) | 0.0 | N/A | N/A | N/A | N/A |
| | | 1.0 | 1 | 1 | 0 | 0 |
| | | 2.0 | 23 | 3 | 18 | 7 |
| | | 3.0 | 28 | 28 | 16 | 16 |
| Breakout | Proposed (ADP) | 0.0 | N/A | N/A | N/A | N/A |
| | | 1.0 | 10 | 4 | 11 | 5 |
| | | 1.5 | 10 | 4 | 11 | 5 |
| | | 2.0 | 10 | 4 | 11 | 5 |

Table 4: Our proposed method with ADP. The max value of maximum tolerable poisoning size $r_t$ of the action-level robustness for the evaluated environments, certified methods, noise levels, and RL algorithms.

The results demonstrate that the empirical performance of our certified defense significantly exceeds the certified lower bound. This observation aligns with the theoretical framework, which defines the certified lower bound as a guarantee for the worst-case scenario, and it provides a conservative measurement of the robustness against attack.

| Attack (Trajectory-level) | Poisoning Proportion | Est. ECR | Min Cumulative Reward | Certified Lower Bound on ECR ($\underline{J}_r$) |
| --- | --- | --- | --- | --- |
| Random Reward | 10% | 79.73 ± 0.44 | 76.52 | 48.76 |
| Random Reward | 20% | 75.94 ± 0.74 | 71.96 | 23.17 |
| Adversarial Reward | 10% | 68.49 ± 0.93 | 61.33 | 48.76 |
| Adversarial Reward | 20% | 60.97 ± 0.58 | 56.39 | 23.17 |

Table 5: Trajectory-level defence with RDP and noise $\sigma = 2.0$ against empirical attacks in the game Halfcheetha.

