# OpenReview forum: "Multi-level Certified Defense Against Poisoning Attacks in Offline Reinforcement Learning"
_ICLR.cc/2025/Conference — ICLR 2025 Poster_

### Official Review · Reviewer_4YPC · 2024-10-18

**Soundness:** 3
**Presentation:** 3
**Contribution:** 3
**Rating:** 8
**Confidence:** 4

**Summary:**

The authors propose a new certified defense technique against poisoning attacks in offline reinforcement learning using differential privacy. The goal of their defense is to certifiably minimize the trained agent's drop in episodic return (policy level robustness), and minimize the probability that one of the agent's actions are changed (action level robustness) when trained on a poisoned dataset $\tilde{\mathcal{D}}$. They further want to extend these certifications to both attackers that modify whole trajectories (trajectory-level poisoning) or individual transitions (transition-level poisoning). They model the strength of the attacker as the total number of alterations they make to the dataset.

Their method, in line with common practice in differential privacy, adds additional random noise to the training of an RL agent given the (potentially poisoned) training data $\tilde{\mathcal{D}}$. They show how their particular construction, given established properties of differential privacy, is able to put a lower bound on the agent's drop in episodic return and an upper bound on how many of the agent's actions are changed. The paper's main contributions are theoretical.

They evaluate their defense against trajectory and transition level attackers of varying strength given discrete and continuous offline datasets. They compare their results against the prior COPA defense, displaying increased episodic return and a larger certified radius.

**Strengths:**

* Defense is well motivated within the scope of prior work.
* Theoretical results are strong and proofs provided in the appendix are convincing.
* The proposed defense improves over prior defenses.
* The writing and presentation is organized and easy to follow.

**Weaknesses:**

My main complaint with this paper is the lack of clarity in the experimental results section. The authors evaluate each defense against different thresholds of $\bar{r}$, but it's very unclear to me what this means. Under the authors definitions of $\mathcal{B}$ the value of $\bar{r}$ is the number of "changes" to trajectories or transitions. Given this definition I could have $\bar{r} = |\mathcal{D}|$ by simply shaping the rewards with some function $F(s,a,s') = \gamma \Phi(s') - \Phi(s)$, which does not alter the optimal policy in an MDP [1]. This would be a "full strength" attack, which manipulates every timestep in the dataset, but it would not actually impact the optimal policy. On the other hand I could invert each reward as $\hat{r}_t = -r_t$, which would have the same "strength" $\bar{r} = |\mathcal{D}|$, but would be a much more impactful attack.

In short, I think the authors need to be more clear about how they're evaluating their defense so readers can have a better understanding of their empirical results. It's also possible that I'm missing some details here, but it seems that $\bar{r}$ is not well defined. I request that the authors:

* Provide a precise definition of how $\bar{r}$ is measured in practice for both trajectory and transition-level poisoning.
* Clarify whether they are actually poisoning the datasets or just evaluating the certifications at different theoretical poisoning levels.
* If actual poisoning is performed, describe the specific attack method used.

I also think the authors should have some discussion on why they do not explicitly consider backdoor attacks. One paper they cite frequently is [2], which is one of the first backdoor poisoning attacks in deep reinforcement learning. The goal of backdoor attacks is to maintain benign episodic return of the agent while inducing adversarial behavior in the agent upon observing a predetermined trigger [3, 4]. I'm not certain, but I believe stronger backdoor attacks like [3] may break your action level robustness without a very large $\sigma$. Therefore, I think motivating why the authors don't explicitly consider backdoor attacks is important (perhaps leave such analysis to a future work?). In summary, I ask that the authors:

* Explicitly discuss why backdoor attacks are not considered in the current work.
* Address whether their certification framework could potentially be extended to cover backdoor attacks, or explain why it may not be applicable.
* Consider adding a discussion on the limitations of their current approach with respect to more sophisticated attacks like backdoors, potentially as part of the future work section.


[1] "Policy Invariance Under Reward Transformations: Theory and Application to Reward Shaping" https://dl.acm.org/doi/10.5555/645528.657613

[2] "TrojDRL: Trojan Attacks on Deep Reinforcement Learning Agents" https://arxiv.org/abs/1903.06638

[3] "SleeperNets: Universal Backdoor Poisoning Attacks Against Reinforcement Learning Agents" https://arxiv.org/abs/2405.20539

[4] "BAFFLE: Hiding Backdoors in Offline Reinforcement Learning Datasets" https://arxiv.org/abs/2210.04688

**Questions:**

* How do you measure attack performance with respect to $\bar{r}$ do you directly poison datasets with some attack and evaluate the relevant metrics or is it something else? If you are performing an attack, which attack are you using or how is it formulated?
* How do you think your defense would work in the presence of backdoor attacks? Do these attacks fit within your certifications?

---

> ### Author Response · Authors · 2024-11-19
>
> We thank the reviewer for their assessment of our paper.
>
> 1. Yes, it is true that an attacker could manipulate the trajectories in a manner that does not influence the resultant policies. However, for certifications, we are concerned about quantifying the impact of a *worst-case* perturbation to the trajectory set.
>
>     As such, when we evaluate against $r$, we are not measuring the effectiveness against any empirical attack of size $r$, but rather calculating the guaranteed resistance of the certified defence to the worst possible perturbation of size $r$.
>
>     In the experiments, we adhere to standard procedures in certified robustness papers by providing a theoretical certification calculated based on training on a *clean dataset* $D$. The certification of any given size $r$ (transition or trajectory-level) is calculated *theoretically* measuring the resistance to the worst possible poisoning attack bounded by $r$.
>
>     While this is discussed in Section 3.2, we have added some additional language here as in lines 175-179, and within the experimental section lines 377-379 to help further clarify this point.
>
> 2. Regarding the concern about **backdoor attacks**, we would like to clarify that backdoor attacks were not explicitly discussed because they can be considered as a subset of poisoning attacks (i.e., training-time attacks) with a specific attack objective. They operate under the same threat model as we considered in the paper (Sec 3.1)---a bounded number/fraction of training data can be corrupted. This threat model is applied in the referenced papers [2] Section 5.2, [3] Section 3.1, and [4] Section 3.4) as well.
>
>     Thus, our proposed certified defense is applicable to backdoor attacks, by offering the same action-level robustness (e.g., ensuring actions remain unchanged in triggered states) and policy-level robustness (e.g., guaranteeing that triggering the backdoor function does not degrade the agent's performance below the certified lower bound) for bounded corruptions in the training dataset.
>
> We again thank the reviewer for their valuable feedback.

---

> > ### Comment · Reviewer_4YPC · 2024-11-25
> > **Response to Rebuttal**
> >
> > Thanks for the response, what you say makes sense. I'm not going to hold you all accountable for the standard practice of theoretical evaluations of poisoning bounds, but it may be interesting and add credibility to the effectiveness of this and future works to include evaluations against real attacks.

---

> > > ### Author Response · Authors · 2024-11-29
> > >
> > > While we appreciate the reviewer acknowledging that our paper aligns with community expectations, in the interests of engaging on your point we have constructed additional experiments that we hope will increase the reviewer's confidence in the merits of our paper.
> > >
> > > To assess the performance of certifications relative to trajectory-level attacks, we implemented two attacks against HalfCheetah when defended using RDP at $\sigma = 2.0$, with the results presented below. These attacks include one in which the rewards in a subset of trajectories are replaced with $r'_i \sim \text{Uniform}[-1,1]$ (Random Reward); and one where they are replaced by $r'_i = -r_i$ (Adversarial Reward), following the methodology of [Ye et al. 2023]. For each attack, our experiments were conducted over $150$ runs to provide the estimated expected cumulative reward (Est. ECR) with 95% confidence intervals, and the minimum cumulative reward observed among all the runs. The corresponding policy-level robustness certification (certified lower bound on ECR) $\underline{J}_r$ for each poisoning size 10% ($r=200$) and 20% ($r=400$) are shown as the same as in Figure 3 in the paper.
> > >
> > > The results demonstrate that the empirical performance of our certified defense significantly exceeds the certified lower bound. This observation aligns with the theoretical framework, which defines the certified lower bound as a guarantee for the worst-case scenario, and provides a conservative measurement of the robustness against attack. While we are unable to update the paper under review at this stage, we will include this content in the final camera-ready submission.
> > >
> > > We hope that this has increased your confidence in the value of our paper, and if so, we would ask for you to consider revisiting your score.
> > > | Attack (Trajectory-level) | Poisoning Proportion | Est. ECR         | Min Cumulative Reward | Certified Lower Bound on ECR ($\underline{J}_r$) |
> > > |---------------------------|----------------------|-------------------|-----------------------|--------------------------------------------------|
> > > | Random Reward             | 10%                 | 79.73 ± 0.44      | 76.52                | 48.76                                           |
> > > | Random Reward             | 20%                 | 75.94 ± 0.74      | 71.96                | 23.17                                           |
> > > | Adversarial Reward        | 10%                 | 68.49 ± 0.93      | 61.33                | 48.76                                           |
> > > | Adversarial Reward        | 20%                 | 60.97 ± 0.58      | 56.39                | 23.17                                           |
> > >
> > > [Ye et al. 2023] Chenlu Ye, Rui Yang, Quanquan Gu, and Tong Zhang. Corruption-Robust Offline Reinforcement Learning with General Function Approximation. Advances in Neural Information Processing Systems, 36:36208–36221, December 2023.

---

> > > > ### Comment · Reviewer_4YPC · 2024-11-30
> > > > **Thank you**
> > > >
> > > > Thank you for including these additional experimental results, they're a very welcome surprise to see. I certainly don't want to diminish the importance of theoretical performance bounds for certified defenses - since we always need to consider the worst case in security - but it really helps to have concrete results against attacks like these even if it's not theoretically "worst case". Seeing the discrepancy between the predicted lower bound and the empirical results makes me wonder what the true worst case attack actually is, and how close it gets to the theoretical lower bound you show here. Perhaps an interesting area for future research.
> > > >
> > > > In either case, showing this level of stability under 20% poisoning rate and completely inverted rewards is very impressive. As such I have increased my score.

---

### Official Review · Reviewer_joBX · 2024-11-05

**Soundness:** 2
**Presentation:** 1
**Contribution:** 2
**Rating:** 3
**Confidence:** 4

**Summary:**

The paper proposes a certified defense framework for certifying offline reinforcement learning algorithms against two levels of poisoning attacks, transition-level and trajectory-level. The proposed algorithm leverages differential privacy (DP) techniques; the derived theoretical guarantees regarding per-state action and expected cumulative reward arises from the properties of DP. Compared to prior works, the proposed work can handle continuous action space and stochastic transition. Empirical evaluations demonstrated the effectiveness of the proposed approach.

**Strengths:**

1. The proposed framework extends beyond the previous work and can handle several important variations including continuous action space, stochastic transition, as well as a transition-level poisoning adversary.
2. The authors conducted extensive evaluations and offered concrete discussions and comparisons.

**Weaknesses:**

1) One major issue with the current status of the draft is the lack of clarity. Important technical details are missing, e.g., the authors didn't clarify how DP is used in the framework; I assume it is through applying DP-SGD when training each sub-policy to ensure each of the obtained set of policies is DP, but this is definitely important technical detail that shouldn't be omitted. Moreover, it's confusing when the authors mentioned in line 240 "depending on whether the DP training algorithm provides transition- or trajectory-level guarantees" while not providing any detail re. what exact algorithm offers what guarantee. An outline of the training/aggregation/certification algorithm will significantly improve the readability.
2) Discussion and comparison with prior works are insufficient, which makes it hard to assess the contribution of the proposed work. The paper didn't make it clear re. what's new in this work & what's different from prior works. 1) The notion of policy-level robustness and action-level robustness was already proposed in Wu et al., 2022, then what is new in Sec 3.2; why the highlight of "multi-level" if it's already covered in the prior work? 2) The authors pointed out the limitation of the prior work being restricted to deterministic environment and discrete action space. Algorithmically, what difference in the proposed approach enables it to address these challenges? The comparisons are particularly important but are missing in the draft.
3) In the experiments, the authors compared with the PARL algorithm in Wu et al., 2022, but the prior work proposed two other algorithms TPARL and DPARL which were reported to achieve better results than PARL on some datasets. It is unclear how the proposed algorithm compares with the two stronger algorithms in the prior work.

Minors:

- Lack of mathematical rigor & typos in various places.
- Line 317: a value network wasn't introduced in the previous context.
- Line 373: a brief description of SimuEM method would help.
- Line 377: it's unclear how "a binary search over the domain of $\mathcal{K}$" Is conducted.

**References**

Wu, Fan, et al. "COPA: Certifying Robust Policies for Offline Reinforcement Learning against Poisoning Attacks." *International Conference on Learning Representations* (2022).

**Questions:**

I have listed in the "Weaknesses" section some major questions re. clarifications of algorithmic details and comparisons with prior works on specific aspects. Below are some additional questions.

1. Why did the authors choose to consider ADP and RDP? How do different versions of DP impact your methods? Is it through the DP accountant used in training the model + the certification results? -- this needs clarification. There exist other advanced and tighter DP accountants, e.g., Gopi et al., 2021, Doroshenko et al., 2022. Can the authors discuss the applicability of these techniques in your work, and whether this will likely strengthen your results?
2. From the experimental results, it seems that RDP almost dominates ADP everywhere. Can the authors provide some insights re. why this happens?

**References**

Gopi, Sivakanth, Yin Tat Lee, and Lukas Wutschitz. "Numerical composition of differential privacy." *Advances in Neural Information Processing Systems* 34 (2021): 11631-11642.

Doroshenko, Vadym, et al. "Connect the Dots: Tighter Discrete Approximations of Privacy Loss Distributions." Proceedings on Privacy Enhancing Technologies (2022).

---

> ### Author Response · Authors · 2024-11-19
>
> We thank the reviewer for their assessment of our paper.
>
> 1. DP is employed through the SGM or DP-FEDAVG mechanisms and used as the basis for our guarantees, as noted in lines 236-241 and detailed in Appendix A.2. This includes our discussion about the algorithms for achieving transition- and trajectory-level guarantees respectively and provides the pseudo-code (Algorithm 1 and Algorithm 2) for implementation details. To help clarify this, we have added additional discussion on this in lines 236-241.
>
>
> 2. Relative to **Wu et al., 2022**,
>     - As for the first concern, Wu et al.'s robustness criteria can only be applied to deterministic environments with discrete actions, while our new robustness criteria extend applicability to stochastic environments and those with continuous action spaces.
>
>         Moreover, the proposed policy-level robustness is novel and different from the ''Lower Bound of Cumulative Reward" in Wu et al. As discussed in lines 118-124 and 506-520, the lower bound in Wu et al. is defined as the minimum *cumulative reward* of a set of specific trajectories. Instead, the policy-level robustness (lines 180-187) is the lower bound of the policy's primary metric---*expected cumulative reward*, measuring the overall performance over all possible trajectories induced by the given policy.
>
>         Finally, Wu et al's criteria only covers trajectory-level poisoning, while ours covers both trajectory- and transition-level. Therefore, we highlight the *multi-level* robustness.
>
>     - Regarding the second concern, the limitations of Wu et al. are due to the Deep Partition Aggregation and tree-search methods they used, which essentially exhaustively iterate through all possible trajectories in a deterministic and discrete setting to find the minimum cumulative reward, as detailed in lines 118-124. Instead, our approach is based on differential privacy (Sec 4.2) which has no such limitation.
>
> 3. Regarding the variants of **PARL**, we would like to note that the action-level robustness of PARL, TPARL, and DPARL are effectively interchangeable (less than 0.05 gap in stability ratio for a given radius)
> as reported in their paper Sec 5.1, Figure 1 (PARL actually achieves slightly better performance in Breakout than the others). We conducted a direct comparison with PARL on action-level under the same experiment setting (Freeway/Breakout using DQN/C51 with 50 partitions) and achieved significant improvement as shown in Figure 1. Considering the similar performance across the variants in Wu et al., we believe it is evident that our method outperforms all variants. We have added more discussion in lines 447-450.
>
>     For policy-level robustness, comparisons are conducted implicitly due to differences in robustness criteria. The advantages of our method as discussed in lines 506-520 are applicable to all the variants in Wu et al.
>
> 4. In responding to the minors
>     - It would be much appreciated if the mathematical rigor and typos could be specified and we will fix that in the revision.
>     - The `$Q_{\tilde{\pi}_i}$` represents the standard definition of action-value in RL as `$\mathbb{E}_{\tilde{\pi}_i} [ \sum_{t=0}^{H-1} \gamma^t r_{t+1} \mid s_0 = s_t, a_0 = a_l]$`. We have added details relating to this point in line 316.
>     - We have clarified the SimuEM method more in lines 364-373 and detailed in Appendix A.5.
>     - The domain of $\mathcal{K}$ consists of a set of parameters as outlined in Definition 3.5. For a given value of $\alpha$ (or $\delta$), the corresponding $\epsilon$ can be calculated as detailed in Appendix A.2. The binary search searches through a given range of $\alpha$ (or $\delta$) with the corresponding calculated $\epsilon$ to identify the parameter pairs that yield the largest $r$ subject to Eq 13. We have clarified it more in lines 364-373 and detailed in Appendix A.5.
>
> 5. For the **choice of DP mechanisms**, we selected ADP as it is the basic and widely used format in DP research, and most DP mechanisms provide privacy bound in Eq 15, enabling the wide applicability of our approach.
> RDP is chosen because of its tighter privacy quantification through sequential function composition, making it popular in deep learning contexts (lines 770-773). We have clarified it more in lines 766-775.
>
>     More advanced DP mechanisms are applicable to the defense through our certification framework as discussed in lines 211-221. For instance, the method in Gopi et al. offers precise privacy loss computation in DP-SGD, which could strengthen our transition-level algorithm. We have added additional discussion on this in lines 861-867.
>
> 6. As discussed, ADP is not as suitable as RDP in managing iterative function composition within deep networks (lines 506-507 and 770-773). Therefore, a tighter accounting of privacy leads to more precise bounds and stronger robustness certification.
>
> We again thank the reviewer for their valuable feedback.

---

> > ### Author Response · Authors · 2024-11-29
> >
> > We thank you for your comments - while we believe we have addressed each of these in our response 10 days ago, we would like to emphasize that we have uploaded a rebuttal revision, which we believe improves the clarity of our work in response to your comments. If there are any other outstanding points we would appreciate the opportunity to discuss those with you. However, if we have resolved your concerns we would appreciate your consideration regarding an updated score.

---

### Official Review · Reviewer_UiiB · 2024-11-05

**Soundness:** 3
**Presentation:** 2
**Contribution:** 4
**Rating:** 8
**Confidence:** 2

**Summary:**

This paper studies the problem of robust DRL algorithms against data poisoning attacks. Based on the idea from differential privacy, it provides a provably robust framework such that by combining an RL algorithm with the framework, one can get a new learning algorithm with a high-probability guarantee on the lower bound of the performance of the learned policy under the attack with a certain level. Compared to the previous work COPA, the performance of the method in this work increases significantly: the performance drops to no more than 50%, with up to 7% of the training data poisoned. At the same time, the percentage COPA can guarantee is only 0.008%. In some sense, this paper develops the first practical and provably robust DRL framework.

**Strengths:**

1. This paper studies an important problem. The threat of poisoning attacks in the real world should not be underestimated. It is crucial to develop methods that are robust against such attacks.

2. This paper provides a framework that can make a class of RL algorithms robust, which is much more powerful than providing a single robust algorithm

3. This paper provides strong theoretical guarantees on its learning framework instead of empirical evaluation, which is especially important for developing trustworthy algorithms.

4. It is interesting to see how ideas from differential privacy help develop a provably robust algorithm in the DRL setting.

5. This paper provides the first practical DRL algorithm/framework against poisoning attacks. The previous work, COPA, had poor performance, which is hard to be realistic about. The experiments result show that the method in this work has a much better performance.

**Weaknesses:**

1. This work does not investigate the lower bound of the guarantee one can get for a robust DRL algorithm under the poisoning attack. So, it is unknown if the method in this work is optimal and the gap between them.

2. The attack's power is too great and unrealistic. The learning algorithm assumes that the attacker can modify some trajectories arbitrarily without any constraint, which may not be realistic for the attack. Currently, the method expects a 50% drop in performance for robustness against 7% of data corruption, which costs too much and can be a consequence of the attack being unrealistic.

3. It is unclear exactly what will happen to the robust algorithm under some practical attacks. The bound provided in the theoretical analysis might not be tight, especially for the practical attacks that do not lead to the worst case. It will be great to see what is the performance and behavior of your robust algorithms under actual attacks.

4. The discussion on the application of differential privacy in RL is missing in the related works

**Questions:**

1. The legends in the figures are missing. They are only explained in the caption.

2. It is important to give a good method a good name. However, it is unclear to me what we should call the method in this work. Could you consider an official name for your method?

---

> ### Author Response · Authors · 2024-11-19
>
> We thank the reviewer for their assessment of our paper.
>
> 1. Regarding the **optimality** of our proposed certified defense, we would like to note that the optimality of our certification relies on the optimality of the DP mechanisms. Research efforts [1,2,3] focusing on optimality in DP could serve as valuable references, and integrating these methods for optimal certification presents an interesting direction for future work. However, we also want to address that the focus of our work is providing a general framework for leveraging DP in certified defenses of offline RL, which enables the application and optimal analysis of more advanced DP mechanisms in the future. We have added additional discussion of this in lines 861-867.
>
> 2. As for the **attacker's power**, we believe the strength of this threat model is equivalent to the assumptions of white-box access in traditional adversarial ML research. In offline RL, a set of trajectories may not be collected immediately by the system as discussed in lines 97-105, 152-157 --- it may be collected in advance, stored, and eventually transferred to the RL learner, which introduces more opportunities for an attacker to compromise the training dataset.
>
> 3. For the concern on **practical attacks**, we emphasize that the current evaluation follows from traditional certification research, where the model is trained on a clean dataset and the certification gives us a measure of what will happen under the worst possible attack. Empirical attacks confined within the radius $\mathcal{B}(D,r)$ will not result in an impact that is larger than the certified bound.
>
> 4. We have added a discussion of the application of DP to RL in lines 791-795.
>
> 5. We have added the legends to the figures.
>
> 6. We would like to give an acronym for our **Mu**lti-level **C**ertified **D**efence as MuCD, and have updated it in the paper.
>
> We greatly value your insights and feedback.
>
> [1] Q. Geng and P. Viswanath, "The optimal mechanism in differential privacy," 2014 IEEE International Symposium on Information Theory, Honolulu, HI, USA, 2014, pp. 2371-2375, doi: 10.1109/ISIT.2014.6875258.
> keywords: {Privacy;Probability distribution;Noise;Data privacy;Laplace equations;Probability density function;Databases}
>
> [2] Q. Geng and P. Viswanath, "The Optimal Noise-Adding Mechanism in Differential Privacy," in IEEE Transactions on Information Theory, vol. 62, no. 2, pp. 925-951, Feb. 2016, doi: 10.1109/TIT.2015.2504967. keywords: {Privacy;Data privacy;Sensitivity;Databases;Laplace equations;Probability distribution;Context;Data privacy;randomized algorithm;Data privacy;randomized algorithm}
>
> [3] Murtagh, Jack, and Salil Vadhan. "The complexity of computing the optimal composition of differential privacy." In Theory of Cryptography Conference, pp. 157-175. Berlin, Heidelberg: Springer Berlin Heidelberg, 2015.

---

### Official Review · Reviewer_pBsV · 2024-11-07

**Soundness:** 4
**Presentation:** 4
**Contribution:** 3
**Rating:** 8
**Confidence:** 3

**Summary:**

The authors investigate certified poisoning defenses in offline RL. The authors employ DP approaches to achieve both action and policy level robustness guarantees.

**Strengths:**

* The paper is very well presented. As someone who is not very familiar with RL poisoning, or certified defenses to poisoning, I think the exposition and organization in the paper was great. The flow was logical and the writing, definitions, and lemmas/theorems were all clear.
* The experimentation is thorough and provides reasonable empirical justification for the proposed method.

**Weaknesses:**

* Although the proposed method does offer transition-level certified robustness, it seems like the trajectory level results presented in Table 1 do not always show an improvement from the proposed method. I'm specifically looking at the Breakout results. I do acknowledge that the proposed method also works for continuous action spaces, and has other flexibility advantages, so I do not consider the previously mentioned results to be of great concern.
* The need to train multiple policies could prove cumbersome in certain settings, and may dissuade practitioners from adopting this approach.

**Questions:**

I'm curious to know how much of a threat existing attacks against offline RL systems actually pose. I know this isn't the focus of this work, but it would be nice to have a bit of explanation about attack specifics/provide more grounding as to why practitioners should be concerned beyond the cited Kumar et al. survey (which I'm not sure is specific to RL poisoning.)

---

> ### Author Response · Authors · 2024-11-19
>
> We thank the reviewer for their assessment of our paper.
>
> - With regard to the **results in Table 1**, we would like to clarify that a fair comparison between our method and COPA should be conducted under a noise level that yields a similar performance (Avg. Cumulative Reward) in a benign environment as discussed in lines 447-451, specifically this corresponds to a noise level of $2$ in Freeway and $1.5$ in Breakout. When the average cumulative reward is controlled for, our technique significantly improves upon the reference cases, demonstrating the utility of our approach. Additionally, Figure 1 provides a more comprehensive comparison with COPA by presenting the full distribution of radii, rather than the mean values shown in Table 1, thereby offering a clearer illustration of the improvements achieved by our method. We have added more clarification to this part in lines 449-452.
>
>
> - Regarding the concern about **training multiple policies**, we acknowledge that this requirement introduces some overhead compared to standard training processes. However, our randomized training process can be executed on a subset of the dataset $D_{sub} \subseteq D$, which enhances efficiency while still ensuring certification for $D$, as indicated in lines 234-235 and further detailed in Appendix A.2.
>
> - Our threat model follows from Kumar et. al., which we believe is indicative of potential risks against ML. While we do not have examples of these attacks being deployed in the wild, we believe that security by obscurity is not a viable defensive strategy for those deploying RL. As such, our intent in this work is to elucidate how the mechanisms of certification can be extended to RL, in order to guard against viable threats. The practice of the attack could be conducted in a manner as we discussed in lines 152-157, 162-166.
>
> Once again, we appreciate your valuable insights and feedback.

---

### Meta-Review · Area_Chair_mdPj · 2024-12-23

**Metareview:**

The paper proposes a certified defense against data poisoning attacks for offline RL agents. The key technique used by this paper is differential privacy, and this technique is novel in bringing certified guarantees to the offline RL setting. The key concerns from the reviews (particularly reviewer joBX) were the relationship with the prior work Wu et al. 2022, and whether the contribution is valid compared to this prior work. Unfortunately, reviewer joBX did not respond to the author's rebuttal and did not discuss the paper with the AC during the discussion period. The AC has carefully read the paper as well as the reviews and author's responses and believes the concerns by reviewer joBX have been addressed, and the methodology discussed in this paper is indeed quite different from Wu et al. 2020. The prior work applied a deterministic certification method, while this work provides a probabilistic guarantee via differential privacy angles, and both approaches are valid and valuable to this field. All other reviewers clearly support the acceptance of the paper, so the final recommendation of the paper is Accept.

**Additional Comments On Reviewer Discussion:**

Reviewer joBX did not join the discussion, while all the other three reviewers reached the consensus of accepting this paper. The AC has also read the paper and reviews and believes the work quality is good enough for acceptance.

---

### Decision · Program_Chairs · 2025-01-22

Accept (Poster)